# Effects of Shift Work on the Eating Behavior of Police Officers on Patrol

**DOI:** 10.3390/nu12040999

**Published:** 2020-04-04

**Authors:** Anastasi Kosmadopoulos, Laura Kervezee, Philippe Boudreau, Fernando Gonzales-Aste, Nina Vujovic, Frank A. J. L. Scheer, Diane B. Boivin

**Affiliations:** 1Centre for Study and Treatment of Circadian Rhythms, Douglas Mental Health University Institute, Department of Psychiatry, McGill University, Montreal, QC H4H 1R3, Canada; anastasi.kosmadopoulos@douglas.mcgill.ca (A.K.); l.kervezee@lumc.nl (L.K.); philippe.boudreau@douglas.mcgill.ca (P.B.); fernando.gonzalesaste@mail.mcgill.ca (F.G.-A.); 2Laboratory for Neurophysiology, Department of Cell and Chemical Biology, Leiden University Medical Center, 2333 ZC Leiden, The Netherlands; 3Medical Chronobiology Program, Division of Sleep and Circadian Disorders, Departments of Medicine and Neurology, Brigham and Women’s Hospital, Boston, MA 02115, USA; nvujovic@bwh.harvard.edu (N.V.); fscheer@bwh.harvard.edu (F.A.J.L.S.); 4Division of Sleep Medicine, Department of Medicine, Harvard Medical School, Boston, MA 02115, USA

**Keywords:** dietary intake, caloric intake, macronutrients, eating patterns, food timing, food choice, chronobiology, chrononutrition, shift work

## Abstract

Recent studies indicate that the timing of food intake can significantly affect metabolism and weight management. Workers operating at atypical times of the 24-h day are at risk of disturbed feeding patterns. Given the increased risk of weight gain, obesity and metabolic syndrome in shift working populations, further research is required to understand whether their eating behavior could contribute to these increased metabolic risks. The objective of this study was to characterize the dietary patterns of police officers across different types of shifts in their natural environments. Thirty-one police officers (six women; aged 32.1 ± 5.4 years, mean ± SD) from the province of Quebec, Canada, participated in a 28- to 35-day study, comprising 9- to 12-h morning, evening, and night shifts alternating with rest days. Sleep and work patterns were recorded with actigraphy and diaries. For at least 24 h during each type of work day and rest day, participants logged nutrient intake by timestamped photographs on smartphones. Macronutrient composition and caloric content were estimated by registered dieticians using the Nutrition Data System for Research database. Data were analyzed with linear mixed effects models and circular ANOVA. More calories were consumed relative to individual metabolic requirements on rest days than both evening- and night-shift days (*p* = 0.001), largely sourced from increased fat (*p* = 0.004) and carbohydrate (trend, *p* = 0.064) intake. Regardless, the proportions of calories from carbohydrates, fat, and protein did not differ significantly between days. More calories were consumed during the night, between 2300 h and 0600 h, on night-shift days than any other days (*p* < 0.001). Caloric intake occurred significantly later for night-shift days (2308 h ± 0114 h, circular mean ± SD) than for rest days (1525 h ± 0029 h; *p* < 0.01) and was dispersed across a longer eating window (13.9 h ± 3.1 h vs. 11.3 h ± 1.8 h, mean ± SD). As macronutrient proportions were similar and caloric intake was lower, the finding of later meals times on night-shift days versus rest days is consistent with emerging hypotheses that implicate the biological timing of food intake—rather than its quantity or composition—as the differentiating dietary factor in shift worker health.

## 1. Introduction

Around 16–30% of the workforce in industrialized countries of North America, Europe, Australasia and Asia regularly engage in some form of shift work, which involves working outside standard daytime hours [1,2,3,4,5]. While shift work is necessary to meet many of the round-the-clock requirements of a functional and increasingly global 24-h society, it is not without cost. In addition to its established effects on performance and fatigue, increasing evidence points to shift work as a health risk [6,7,8,9,10]; indeed, shift work is associated with greater likelihoods of obesity [11], hypertension [12], diabetes [13], and cardiovascular disease [14]. Though many behavioral, environmental, and physiological factors contribute to these conditions, the unifying factor in shift work is the disruption of the circadian system—caused by the disassociation of regular behavioral cycles (e.g., sleep–wakefulness, rest–activity, fasting–feeding) from the rhythm of the circadian pacemaker, i.e., the “body clock” [8,15,16,17,18,19]. Circadian disruption and sleep disturbance that arise from this misalignment have both been shown to impair energy metabolism, glucose metabolism, and the production of appetite-regulating hormones, which can lead to weight gain and the development of metabolic disorders [20,21,22,23,24,25,26].

Considering the significance and prevalence of adverse health outcomes in shift work, there is desire from a public health perspective to identify and better understand suitable targets for intervention. As a modifiable behavior that affects energy balance and the synchronization of metabolite rhythms, food consumption is a plausible candidate for mitigating disease risk [15,27,28,29,30]. Considerable information about the food choices of shift workers has been gained in various studies via questionnaires, interviews, focus groups, diaries, and 24-h recall [17,27,31]. Many of these studies report either no or limited differences in total daily caloric intake between shift types, or between shift workers and non-shift workers [32,33,34,35,36,37,38,39]. However, there is some evidence that caloric intake may differ between rest days and work days [32], and that engaging in shift work affects the macronutrient composition of meals [31,39]. Indeed, working evening or night shifts is variously described in the literature to be associated with consuming less protein, and/or more carbohydrates, total sugars, fats, or alcohol [37,38,39,40,41,42]. One of the clearest dietary impacts of shift work is the overall displacement of the fasting/feeding cycle and its effect on meal timings across the day. The primary findings of studies which describe the temporal distribution of food consumption indicate that shift work is associated with more disparate meals, fewer and smaller breakfasts, later meal times, and increased caloric intake at night [32,37,39,43].

While portion size and macronutrients are important factors involved in nutrition, there is growing interest in the role that food timing has on health due to its influence on physiological processes that affect metabolism [9,15]. In previous studies of non-shift working individuals, eating food at night or late in the day has been shown to be associated with impaired glucose metabolism, increased total caloric intake, and reduced weight loss effectiveness [23,44,45,46,47,48]. Indeed, experimentally delaying evening meal timing causes an impairment of glucose tolerance [49]. Other experiments have suggested that changes in behavioral cycles, including those of fasting/feeding, induce circadian misalignment of peripheral clocks in the liver, pancreas, and gastrointestinal tract which may partially explain the resultant impairment of metabolic processes [50,51]. In recent years, several research groups have implemented a novel approach for investigating dietary patterns involving a smartphone-based application to log meals with timestamped photographs [52,53,54,55,56]. Using this method, Gill and Panda [52] found that half of the adults in their study consumed meals over a period of 15 h or longer per day, with a bias towards consuming the majority of calories later in the evening. Importantly, they also observed that overweight individuals who restricted meals to a 10–11-h daytime window experienced substantial and persistent weight loss, raising the profile of time-restricted eating (TRE) as a potential therapeutic approach for improving health [52]. Similar benefits of TRE were recently observed in patients with metabolic syndrome using the same methods [54]. Several additional studies with young, non-shift-working adults further supported the usefulness of this record-keeping methodology. For instance, McHill et al. demonstrated that high body fat is associated with eating at later circadian phases rather than later times of day [55], and that high body fat is associated with greater caloric intake at later circadian phases [56]. 

The use of this smartphone-based method for characterizing food intake patterns is ideal for long-term monitoring of nutrition because it is less intrusive of daily activities than other methods of record-keeping and it provides objective information about meal content and timing, suffering less from recall biases than questionnaires [27]. Its successful implementation in the aforementioned studies highlights its potential for use with other populations of interest [52,53,54,55,56]. However, to the best of our knowledge, it has yet to be employed with a shift-working population. While useful data about shift worker diets have been collected using other methods, relatively few studies have been able to characterize the profiles of food consumption across multiple different shift/rest days with timestamps in the same individuals [31,39]. This information is important for identifying potential widespread behaviors requiring intervention [27]. 

Police officers are a class of shift worker that may benefit from dietary intervention to improve their health. Police officers have been shown to have high rates of obstructive sleep apnea, overweight and obesity, and diabetes [57]. In another study, police officers on 12-h shift rosters were shown to have elevated systolic blood pressure that increased at the end of their shift [58]. In this study, we describe the dietary patterns and caloric intake of police officers on patrol across four different types of day during their work cycle—namely, a rest day in which they did not work and days during which they had a morning shift, an evening shift, and a night shift. During these days, participating police officers logged all their meals and beverages with timestamped photographs using a mobile phone. Thus, we could reliably assess how their work schedules affected the timing and composition of their meals.

## 2. Materials and Methods 

### 2.1. Participants

Police officers, aged 20–67 years, working full-time on rotating shifts in the province of Quebec, Canada were contacted between September 2016 and December 2017 via official email channels following approval of both management and union representatives. Police officers who expressed interest were required to meet eligibility criteria during a phone-based screening questionnaire. Prospective participants were excluded if they regularly engaged in substantial commitments other than their police patrol duties in excess of 10 hours/week; had a medical condition or were prescribed medication that could disturb sleep or waking; or reported sleep disorders unrelated to their rotating work schedules. Those participants eligible based on these criteria underwent additional screening, including a physical assessment by a physician and assessment of blood and urine samples. Through this process, anthropometric data were collected, including the body mass index (BMI; kg/m^2^), waist-to-thigh ratio (WTR), waist-to-hip ratio (WHR), and estimated basal metabolic rate (BMR) [59]. Participants also completed the Munich ChronoType Questionnaire for Shift Workers (MCTQ^Shift^) to derive a chronotype estimate from mid-sleep on rest days (MSF) [30]. 

A total of 81 out of 159 police officers who responded to the advertisement were eligible and expressed interest in participating in a larger field study. Of these, a subset of 53 officers was invited to participate in the present food study; 36 agreed to do so (Appendix A). From these, 31 officers contributed to the final dataset, as 5 had to be excluded due to dropout (*n* = 1), technical issues with equipment (*n* = 2) or withdrawal for medical reasons (*n* = 2).

#### Ethics

Ethical approval for the study was granted by the Douglas Mental Health University Institute Research Ethics Board (IUSMD-14-15). Participants were informed of their ability to withdraw from the study at any time and provided written informed consent. The study was conducted in accordance with the Declaration of Helsinki.

### 2.2. Study Procedures

Police officers participated in an observational field study in which they documented their sleep, work, and eating behavior whilst working their habitual rosters (Figure 1A). Participants provided their work rosters to the research team in advance of the study to facilitate planning. Of the 31 police officers contributing to the study, 27 habitually worked a 35-day cycle with a roster comprising series of 9- and 12-h morning/day shifts (0700–1600 h, 0700–1900 h), evening shifts (1500–2400 h), and night shifts (2230–0730 h, 2300–0800 h, or 1900–0700 h) separated by rest days (Appendix A). The remaining four police officers worked a 28-day cycle with a roster comprising 12-h day shifts (0700–1900 h) and 12-h night shifts (1900–0700 h) separated by rest days (Appendix A).

Participants were provided a smartphone (Nexus 5, LG Electronics, Seoul, Korea) and instructed to use it to photograph all food and beverage intakes on days pre-selected by the research team following examination of their work schedules. A staff member reminded participants to begin logging meals on the day prior to each nutrient collection period. For participants on the 35-day roster, these days encompassed the second rest day and the second morning, evening, and night shift in a sequence—i.e., four “day types” in total (Figure 1A, Appendix A). For participants on the 28-day roster, these days encompassed the second rest day and the second day shift and night shift in a sequence—i.e., three “day types” in total (Appendix A). For each selected day type, the nutrient analysis window began from the offset (i.e., wake time) of the main sleep period prior to the shift/rest day until the offset of the main sleep period that followed. After taking a photograph, participants were instructed to classify the meal type—e.g., “breakfast”, “lunch”, “dinner”, “snack”, or “beverage only”—using the movisensXS smartphone application (Movisens GmbH, Karlsruhe, Germany). If participants did not finish a meal, they were asked to take a second photograph of the remainder. Finally, at the end of each day, police officers were asked to complete a chart describing the type and quantity of any food supplements they had consumed (e.g., vitamins).

Throughout the work cycles, police officers wore an actigraph (Actiwatch Spectrum^®^, Respironics, Philips, OR) that comprised an accelerometer, light sensor, and event marker on the wrist of their non-dominant hand to monitor sleep–wake and physical activity patterns; they were asked to press the event marker at each bedtime and rise time. The device was required to be worn at all times except when showering, swimming, or engaging in contact sports to avoid equipment damage. Multiple times each day, participants also completed timestamped questionnaires on the smartphone. These questionnaires were used to log their bedtimes, rise times, and work times (i.e., work start, work end), as well as to collect data on neurobehavioral performance and self-reported fatigue, sleepiness, and mood for another study. Participants were contacted weekly throughout the study to limit data loss and could contact a staff member at any time for study assistance. At the conclusion of the study, police officers returned their equipment with a copy of their final work schedules so the research team could cross-check and verify their actual work hours.

### 2.3. Processing of Photographs of Meals and Beverages for Nutritional Content

At the end of the study, a staff member contacted each participant to clarify and confirm the contents of their photographs. Descriptions of the food and beverage items and portion sizes were entered into a spreadsheet; these were then checked for ambiguities by at least one other staff member to ensure that the descriptions were clear. Photographs then underwent anonymization with respect to date, time, and day type. Nutrient analyses were conducted by a registered dietician of the Center of Clinical Investigations at Brigham and Women’s Hospital using the University of Minnesota Nutrition Data System for Research software [60]. The accuracy of entries into the software were verified by another staff member. Among the dietary information extracted from these assessments were total calories (kcal) and macronutrient calories obtained from carbohydrates, fats, and proteins. Consistent with a previously published methodology, meals of the same type (e.g., “snack”, “beverage only”, etc.) that were consumed within 15 min of each other were combined into a single intake event, using the timestamp of the last meal [52,53,54,55,56].

### 2.4. Shift Verification and Classification

To verify the work times, self-reported work hours were cross verified with work schedules provided to the research staff and the timestamps of questionnaires completed at the start and end of the shifts. Any conflicting work-start and work-end times were investigated and corrected. After these had been confirmed, shifts were classified based on their timing. Work periods commencing between 0500 h and 0900 h were classified as morning shifts; those ending between 2200 h and 0200 h were classified as evening shifts; those that encompassed the period between 0100 h and 0500 h were classified as night shifts. To facilitate the comparability between shifts, atypical work periods longer than 13.5 h (i.e., overtime of more than half a 9-h shift) or shorter than 4.5 h (i.e., less than half a 9-h shift) were excluded from analyses. Figure 1B and Appendix A illustrate the number of participants contributing to each combination of valid shifts/rest days. 

### 2.5. Sleep–Wake Assessments

A standardized hierarchical algorithm adapted from a previously published methodology [61] was used to identify participants’ rest intervals (i.e., the period from bedtime to rise time) from sleep log information and timestamps, together with physical activity, event markers, and light exposure data retrieved from the actigraph in 15-s epochs. Once rest intervals had been identified, a sleep detection algorithm was used to define the sleep period onset and final offset (i.e., wake time) and apply a sleep/wake estimate for each 15-s interval between them based on the activity counts in surrounding epochs [62,63]. From these data, the primary measure was sleep duration, defined as the total amount of sleep (in hours) scored within the sleep period. When more than one sleep period occurred within a 24-h calendar day, the longest sleep period was defined as the main sleep and all others were defined as naps. For the purpose of this study, relevant sleep episodes, including naps, were those that enclosed nutrient collection days, between the start of the main sleep immediately prior to a shift/rest day and the wake time of the following main sleep period.

### 2.6. Statistical Analyses

Data analyses and visualization were performed using R Foundation for Statistical Computing, Version 3.6.0 [64] and the packages “ggplot2” and “ggbeeswarm” [65,66]. Demographic and anthropometric measures were compared between sexes with Welch’s T-tests. *Χ*^2^ tests were used to compare the proportion of meal intakes by size, time and location of consumption, and meal type. Analyses of circular data (i.e., clock times), including meal, sleep/wake, and work times, were conducted with Circular ANOVA using the R package “circular” [67]. Remaining analyses of meal frequency and composition, physical activity, and work/sleep durations were conducted with linear mixed-effects models using the R package “lme4” and “lmerTest” [68,69]. These models controlled for sex and age and included “subject ID” as a random effect. Post-hoc pairwise comparisons were made using the package “emmeans” [70]. Significance was determined at *p* < 0.05 for main effects and post-hoc tests, corrected for multiple comparisons using the Benjamini–Hochberg method [71].

#### 2.6.1. Total and Macronutrient Caloric Intake

Since participants had different energy requirements and did not equally contribute to the shift types or rest days included in the final dataset, caloric intake was standardized relative to basal metabolic rate (BMR) using the Mifflin–St Jeor equation [59]. Specifically, the total and macronutrient calories consumed per day type were expressed as percentages of participants’ basal metabolic rate (% BMR). For this purpose, the BMR was weighted for the duration of the ~24-h “day”—from the wake time of the main sleep period before the shift/rest day until the wake time of the following main sleep period (i.e., 24-h BMR × [day length/24]). The total and macronutrient % BMR were assessed using mixed-effects model ANOVAs, with “day type” specified as the fixed effect.

#### 2.6.2. Timing and Spread of Meals

Each meal intake was assigned to both a clock time, as well as an elapsed number of hours after awakening from the main sleep episode, minus the duration of any preceding naps. The interval between the last meal and the next main sleep onset was also calculated. From these, three dependent variables were created: (i) elapsed time awake at the first meal (>5 kcal), (ii) time until sleep from the last meal (>5 kcal), and (iii) the duration of the eating window between first and last meals. The spread of total and macronutrient calories around the clock was assessed by calculating the weighted circular r*ho*-value of meal times (>5 kcal) for each participant [67]. This value represents the mean resultant vector length of circular data, with smaller values indicative of greater spread. The times of meals and macronutrient caloric intakes were weighted by their sizes to produce a circular mean intake time for each participant. Linear mixed-effects models and circular ANOVAs were conducted, with “day type” as the fixed effect.

#### 2.6.3. Effects of Day Type, Time-of-Day and Time Awake

The distribution of caloric intake between and within day types was assessed by converting the content of meal intake into percentages of individuals’ total intake per day type. For day type comparisons, the percentages of total intake from each macronutrient were calculated. Percentages were also calculated for total calories and macronutrients consumed at work (i.e., between work start and work end), and consumed at night (i.e., 2300–0600 h). Linear mixed-effects models specified a single fixed effect of “day type”. 

To assess the distribution of calories by time-of-day and time since waking from the prior main sleep episode (adjusted for naps), percentages of total intake were calculated across 4-h bins for each day type. If a participant did not have a caloric intake during a time-of-day or time since waking bin, or if they did not have a caloric intake at night or at work, a value of zero was assigned. Linear mixed-effects models were used to assess the interaction of “day type” and “time-of-day” and of “day type” and “time awake”.

## 3. Results

### 3.1. Participants

The 31 (6 female) participants included in the analyses had a mean age of 32.1 ± 5.4 years (25–44 years; mean ± SD, range), a mean BMI of 25.0 ± 2.3 kg/m^2^ (20.7–29.9 kg/m^2^), and mean experience in the police force of 8.6 ± 4.7 years (3–21 years) (Table 1). A correlation suggests overweight in this sample is associated with age (*r*(31) = 0.40, *p* = 0.026). Of the other characteristics, the only significant (*p* < 0.05) difference observed between sexes was for their estimated 24-h basal metabolic rate (BMR). As expected, the average BMR of male police officers (1782 ± 121 kcal, 1579–2032 kcal; mean ± SD, range) was higher than that of female police officers (1388 ± 109 kcal, 1238–1562 kcal, *p* < 0.001).

### 3.2. Work Days and Rest Days 

Of the 107 relevant data collection days obtained, 16 were removed due to atypical shifts (i.e., <4.5 h or >13.5 h) and one was removed due to insufficient data (i.e., only one meal documented), resulting in 90 valid days in the final analyses (Appendix A). No participant contributed more than once per day type. A *Χ*^2^ square goodness-of-fit test indicated the distribution of days across rest day (*n* = 28, 31%), morning shift (*n* = 21, 23%), evening shift (*n* = 17, 19%), and night shift (*n* = 24, 27%) day type categories was comparable (*X*^2^ = 2.89, *p* = 0.41).

The mean work start and end times (circular mean) were: 0658–1627 h for morning shifts, 1458–0002 h for evening shifts, and 2153–0751 h for night shifts (Appendix A). Start and end times significantly differed between shift types (*p* < 0.001), but work durations did not (*p* = 0.098). Activity counts from the actigraphs indicate that police officers were more active on rest days than night-shift days (292 ± 79 vs. 251 ± 64 counts/min, *p* = 0.017) (Appendix A). Police officers were also more active at work during morning shifts than they were at work during night shifts (280 ± 52 vs. 230 ± 52 counts/min, *p* = 0.003).

### 3.3. Sleep–Wake Comparisons 

Sleep onset and wake times were significantly affected by day type (Appendix A). Wake times prior to morning shifts (0532 h ± 7 min) were significantly earlier than on rest days (0800 ± 24 min), or prior to evening shifts (0834 h ± 26 min) and night shifts (1558 h ± 39 min). Sleep onset times at the end of the wake period also reflected the work schedules; sleep onsets following night shifts (0914 h ± 17 min) were significantly different from sleep onsets on rest days (2256 h ± 18 min), or following morning shifts (2215 h ± 15 min), and evening shifts (0111 h ± 9 min). 

Day types did not significantly differ in “day length” from wake time of the main sleep period before the shift/rest day until the wake time of the main sleep period that followed (mean range: 23.4–24.6 h, *p* = 0.165). Police officers spent a greater proportion of time asleep on rest days (34%) than work days (27–30%, *p* < 0.05), and more on morning-shift than night-shift days (30% vs. 27%, *p* < 0.05; Appendix A).

### 3.4. Description of Meal Intakes

A total of 569 meal intakes were analyzed, distributed between rest days (*n* = 210, 37%), morning-shift days (*n* = 133, 23%), evening-shift days (*n* = 85, 15%), and night-shift days (*n* = 141, 25%). Since different numbers of participants were included for each day type (Appendix A), a *Χ*^2^ goodness-of-fit test was conducted to test whether the total count of meals documented for each day type was proportionate to the number of contributing participants for each day type. The test indicated that the number of meals significantly differed from expected values (*Χ*^2^ = 11.61, *p* = 0.009). Pairwise comparisons showed more meals than expected were logged on rest days relative to evening-shift days (*Χ*^2^ = 10.09, *p* = 0.004) and night-shift days (*Χ*^2^ = 5.05, *p* = 0.037). 

On average, police officers obtained significantly more meals on rest days (7.5 ± 2.2), and significantly fewer meals on evening-shift days (5.0 ± 1.4, *p* < 0.001), than all other day types (Table 2). In further analyses, *Χ*^2^ tests of independence showed that participants consumed a greater proportion of their meals at night on night-shift days (33%) than they did on other days (1–14%) (Appendix A). However, there were no differences between day types in terms of: the proportion of meals within different caloric ranges; the proportion of intakes classified as “main meals” (48–53%), “snacks” (27–32%), or “beverage only” (16–26%); and the proportion of meals at work (40–47%). The average size of officers’ main meals (39–41% BMR, *p* = 0.84) and snacks (14–18% BMR, *p* = 0.84) did not differ between day types.

### 3.5. Total Caloric and Macronutrient Intake as a percentage of Basal Metabolic Rate

The average estimated 24-h basal metabolic rates of participants contributing to each day type did not differ (Table 2). Participants consumed the greatest percentage of their basal daily energy requirements on rest days and the smallest percentage on evening-shift days (Figure 2). Consumption on night-shift and evening-shift days was significantly lower than on rest days (Table 2). Consumption on evening-shift days was also lower than on morning-shift days. When assessed at a macronutrient level, participants consumed more calories as a percentage of their BMR from fat and saturated fat on rest days than on evening-shift days and night-shift days. Participants also consumed more fat on morning-shift days than evening-shift days. Intake of other macronutrients did not differ between day types, although a trend was observed for carbohydrate intake (*p* = 0.064; Table 2).

### 3.6. Timing and Dispersion of Meals 

On average, the first meal (>5 kcal) occurred between ~1 h to 2 h of elapsed time awake, but there was no significant effect of day type (*p* = 0.203) (Appendix A). Similarly, final meals occurred, on average, between ~2 h to 3 h before sleep onset but did not differ between day types (*p* = 0.303). A significant effect of day type was found for the eating window (F(3,63) = 5.65, *p* = 0.005) (Appendix A). It showed that the duration between first and final meals was significantly longer for night-shift days than rest days (13.86 h ± 3.14 h vs. 11.26 h ± 1.81).

Mixed-effects models showed a significant fixed effect of day type on the weighted *rho*-value used to measure the spread of caloric intake around the clock (F(3,84) = 8.08, *p* < 0.001). It showed that meals were significantly more dispersed on night-shift days than rest days or other work days (Figure 3A). Analyses of the macronutrients showed similar results (Appendix A).

A circular ANOVA used to assess the weighted mean times of meal intakes revealed a significant main effect of day type (F(3,86) = 29.27, *p* < 0.001). Mean caloric intake occurred significantly later on night-shift days (2308 h ± 74 min) than on rest (1525 h ± 29 min), morning-shift (1410 h ± 27 min), and evening-shift (1453 h ± 39 min) days (Figure 3B). Mean intake on morning-shift days was significantly earlier than on rest days. Macronutrient results were similar (Appendix A).

### 3.7. Caloric and Macronutrient Intake by Time-of-Day and Time Awake

Mixed-effects analyses of the percentage of total caloric intake revealed significant interactions of day type with time-of-day for the percentage of total caloric intake (F(15,516) = 7.33, *p* < 0.001), and the percentages of total caloric intake from carbohydrate (F(15,516) = 6.72, *p* < 0.001), fat (F(15,516) = 5.29, *p* < 0.001), and protein (F(15,516) = 4.66, *p* < 0.001) (Figure 4; Appendix A). For all day types, the distribution of caloric intake varied as a function of time-of-day. In each, the greatest proportion of calories (~30–35%) occurred between 1600–2000 h. For rest days and for morning-shift and evening-shift days, the trough of consumption occurred between 0000–0800 h. On morning-shift days, there was also a reduction of caloric intake from 1200–1600 h compared to 1600–2000 h. Night-shift days displayed a less robust diurnal rhythm of caloric intake, with the lowest proportion of total calories shifted to 0800–1600 h. There were similar rhythms at the macronutrient level (Appendix A).

Mixed-effects analyses revealed significant interactions of day type with time awake for the percentage of total caloric intake (F(12,440) = 6.66, *p* < 0.001) and the percentages of total caloric intake from carbohydrate (F(12,440) = 5.60, *p* < 0.001), fat (F(12,440) = 4.95, *p* < 0.001), and protein (F(12,440) = 6.53, *p* < 0.001) (Figure 4; Appendix A). The distribution of total caloric intake remained stable for the first 8 h after waking on rest days, morning-shift days, and evening-shift days (Figure 4). Caloric intake was significantly reduced between 8 to 12 h after waking on morning-shift days, before increasing again 12 to 16 h post-waking. On both evening-shift and night-shift days, the greatest percentage of calories were consumed within the first 4 h post waking. For night-shift days, this was significantly more than consumed at any other time awake (Figure 4). For evening-shift days, this was significantly more than consumed after 12 h of time awake. For all day types, the proportions of total caloric intake significantly tapered beyond 16 h of wakefulness. Similar distributions of caloric intake with time since waking were observed for the macronutrients (Appendix A).

### 3.8. Calories and Composition of Macronutrients Overall, at Night, and at Work

Mixed-effects model ANOVA revealed that the percentages of total calories consumed from carbohydrates, sugars, fats, saturated fats, and proteins did not significantly differ between the day types (Table 3). Overall, the average sources of daily calories were 44% from carbohydrate, 36% from fat, and 18% from proteins.

When comparing the percentages of total calories consumed at night (i.e., 2300–0600 h), a significantly greater percentage was consumed on night-shift days than on all other day types (Table 3). Participants also consumed a greater percentage of total calories at night on evening-shift days than they did on rest days and morning-shift days. 

When comparing the percentages of total caloric intake from all meals or macronutrients at work, there were no significant main effects of day type (Table 3). Post-hoc analyses of the main effect of day type trends (*p* < 0.10) suggested that participants consumed more calories at work during morning shifts than at work during evening and night shifts (*p* < 0.05). Further, post-hoc analyses for the macronutrient trends indicate they consumed more carbohydrates at work during morning shifts than evening shifts (*p* = 0.029), and more fat at work during morning shifts than night shifts (*p* = 0.040) (Table 3).

## 4. Discussion

Shift workers are at increased risk of weight gain and developing diabetes and cardiovascular diseases. These are considered to be a consequence of a multitude of factors, including poorer health behaviors (e.g., diet, exercise) and impaired metabolic function caused by sleep–wake disturbance and circadian misalignment of physiological rhythms [9,10,21,22,23]. While shift work permits little flexibility to alter sleep–wake times, diet is a behavior that may be plausibly modified to mitigate long-term health risks [27]. In the current study, our goal was to objectively characterize the dietary patterns and caloric intake of police officers on a rotating shift schedule, for the purpose of identifying potential targets for intervention on different days of their roster. To achieve this, we employed a novel method of food documentation involving smartphones and dietetic analysis of timestamped photographs. To our knowledge, this is the first time this approach has been used with shift workers.

### 4.1. Sleep and Work Factors in Meal Consumption 

The times at which police officers were able to eat on a given day were contingent on the sleep–wake times and work breaks permitted by their work schedule. Indeed, compared to rest days, officers had to wake up significantly earlier prior to morning shifts and began sleep later at night following evening shifts or in the morning following night shifts. As a consequence, police officers obtained significantly less sleep on work days than rest days as a proportion of the day. Associated with these altered sleep–wake times, police officers consumed their meals earlier on morning-shift days than rest days, and—likely of more importance for metabolic health—at considerably later times of day for night-shift days than other day types. As police officers’ circadian pacemakers were unlikely to have adapted to a nocturnal schedule by the second shift, these later meals would have occurred at later phases of their internal clock. The greatest proportion of daily calories (~30–35%) occurred between 1600–2000 h on all day types—i.e., after morning shifts, during evening shifts, and before night shifts. On morning-shift days, the reduction of caloric intake from 1200–1600 h coincided with the latter half of the shift and the post-work commute.

Another consequence of reduced sleep duration on work days, particularly following night shifts, was that police officers had inversely longer wake times during which they could eat [73]. Indeed, police officers had the shortest eating window (11.3 h ± 1.8 h) on rest days where the most sleep was obtained and had longest eating window (13.9 h ± 3.1 h) on night-shift days where the least sleep was obtained. Similar trends were present for the other work days. While fewer meals were eaten on night-shift days than rest days, this duration in itself is important because extending the eating window reduces the time during which bodies can enter an important fasting state that facilitates cellular repair and activates catabolic processes (e.g., the synthesis of glucose) [74,75]. Moreover, the variability in meal times and eating windows across rest days and work days is of significant concern for shift workers on rotating schedules because irregular eating patterns can also impair metabolic function [76,77].

### 4.2. Meal Frequency, Quantity and Macronutrient Composition

A comparison of the number of meals documented for each day type revealed that more meals were consumed on rest days than evening-shift and night-shift days. However, none of the days significantly differed in terms of the proportion of meals that were snacks or only composed of beverages. As police officers did not contribute to all work days and rest days in the final dataset, we standardized caloric intake during these meals relative to their estimated daily BMR (see methods). This ensured we could evaluate differences in total intake between day types according to their individual requirements. The estimation of the BMR did not significantly differ between the police officers who contributed to the different day types.

Unlike much of the literature about dietary patterns in shift work, we observed that total caloric intake by police officers significantly varied between day types. Specifically, police officers consumed more calories relative to their metabolic requirements on rest days and morning-shift days on average (170%–184% BMR; Table 2) than on evening-shift and night-shift days (130–143% BMR). The increase in total caloric intake on rest days and morning-shift days appears to be due to an increase in calories from fats (*p* = 0.004) and carbohydrates (*p* = 0.064). Gill and Panda estimated that participants forget to log approximately 10% of their food/beverage or water events, meaning that caloric intake is likely higher than reported [52]. However, the false negative rate may be lower in the current study as participants only had to log meals for a single day at a time, rather than multiple weeks, and were reminded by staff the preceding day. Assuming that low to moderate physical activity each day would require a further ~40–60% in caloric intake [78] relative to BMR to account for additional energy expenditure, it is possible police officers overate relative to their metabolic requirements on rest days and morning-shift days and/or underate on evening-shift and night-shift days. However, without information about their actual total energy expenditure on each day, it is not possible to categorically state whether police officers were in positive (or negative) energy balance on any particular day of the study. Accelerometry data from the actigraphs suggest that police officers were most physically active on day-oriented schedules and least active on night-oriented schedules (Appendix A), so it is possible they were able to maintain energy balance. Nonetheless, the consumption differentials between each of the work days and rest days provide useful information about the police officers’ dietary choices and a potential area of focus for weight management.

In contrast to what we observed in our sample, Lennernas et al. found that shift workers consumed more on a 12-h day shift day than on rest days [32]. However, this was a consequence of these shifts always occurring on Sundays, and thus being accompanied by traditional large meals of meat and potatoes. Apart from the 12-h shift, consumption on rest days did not differ from the other morning, evening, or night shifts. It is not clear why there are differences in total caloric intake between day types in the current study, yet relatively small differences in other studies [32,39].

In terms of the overall macronutrient composition of meals, there were no significant differences between day types. On average, the sources of daily calories across all day types were proportionally 44% from carbohydrates, 36% from fats (including 12% saturated fat), and 18% from proteins. These quantities fell slightly short of the recommended levels of carbohydrates (45–65%) and slightly exceeded the recommended levels of both fat (20–35%) and saturated fat (<10%) [78], but are consistent with what has previously been reported in shift workers [39,79].

### 4.3. Total Caloric and Macronutrient Intake at Night and at Work

In other studies, the primary consequence of night work is the increase in meals and caloric intake consumed during the night relative to the daytime [32,39], particularly in the form of snacks [37]. Our results are consistent with these previous findings. In another study of the temporal eating patterns of shift workers, Shaw et al. found that 6% of meals occurred at night when working days, in contrast to 30% when working nights [39]. We found that the proportions of meals at night were similarly related to day type—4.5% for morning shifts, 14.1% for evening shifts, and 32.6% for night shifts (Appendix A). Caloric intake was distributed in a similar fashion, with 1.2%, 9.7%, and 30.3% of total intake consumed from 2300–0600 h at night on morning-shift, evening-shift, and night-shift days, respectively. For perspective, the proportion of calories between these hours on night-shift days, presumed to have been consumed in misalignment with the circadian pacemaker due to the low rate of adaptation, constituted a 14-fold increase relative to the amount consumed in the same period on rest days. Nonetheless, the overwhelming majority (~70%) of calories consumed on night-shift days were still consumed during the day. 

Although the duration of work periods did not differ between shift types (Appendix A), there was a statistical trend (*p* = 0.083) in which police officers tended to have a lower caloric intake during night shifts (43.4%) than during morning shifts (52.5%). This pattern of greater caloric intake during morning shifts than night shifts is consistent with the observations of Lennernas et al. [32]; furthermore, it comports with findings that circadian rhythms of hunger—and its associated hormone, ghrelin—peak in the biological evening and reach a trough during the biological night [24,80,81]. The circadian rhythms of hunger may also explain the marked peaks in caloric intake between 1600–2000 h for all day types, including night-shift days, despite an overall dampening of intake across the wake period on night-shift days.

### 4.4. Strengths, Limitations, and Future Directions

The strengths of the current study are that we were able to obtain information about the diets of police officers across the variety of representative days experienced in their work schedules and did so using a novel smartphone-based approach with timestamped photographs. The advantages of monitoring nutrition in this way are that it provides precise and reliable information about food timing and allows for the objective quantification of the various macronutrients depicted in the photographs by dieticians. Moreover, it is less susceptible than other measures to the limitations of human memory (food content and timing) and the lack of feedback or human interaction make it less prone to social desirability bias (over-reporting positive behaviors and under-reporting negative behaviors), and the observer effect (altering behavior in response to being monitored). The successful implementation of the method in this and other studies [52,53,54,55,56] demonstrates a viable path for the future assessment of long-term dietary patterns in different demographic populations and shift working occupations (e.g., nursing).

While acknowledging these strengths, it is important to consider that this study only captured a snapshot of eating behavior for each shift. As participants did not contribute more than one day of data for each day type, it is not possible to know the extent to which individual eating patterns may vary from day-to-day per shift type. Thus, our results assume data were collected on ‘typical’ days. We tried to ensure this was the case in a couple of ways: first, we selected the second day in sequences of shifts for data collection to avoid transitions between day types. Additionally, we excluded days from the analyses that were unusual, such as those with work periods that were unusually long or short. It is worth emphasizing that the police officers were all shift workers on rotating schedules; we did not compare shift workers with non-shift workers. Thus, it is feasible that the size and composition of meals consumed in our study differ from the habitual dietary patterns of permanent day workers and permanent night workers. 

Furthermore, the police officers in this study are a select group of shift workers and may not represent the majority of shift workers in other occupations (e.g., those that are more sedentary or involve working indoors) whose daily routines may differentially affect their dietary choices/options and physical activity [31]. Indeed, the tendency for greater body mass and cardiovascular symptoms in nurses who work shifts [82,83] may be related to their reports of (i) reduced time and energy to prepare healthy meals, (ii) few options available at night, and (iii) limited opportunities to eat while at work [84,85]. These factors may increase the likelihood of choosing convenient and unhealthy “fast food” options [31]. Compared to shift workers primarily based at their desk or indoors, police officers out on patrol have different work conditions that can affect both accessibility to healthy food and levels of physical activity during the course of their duties.

It should be noted that, as a group, the police officers in our study were healthy, with no medical or psychiatric conditions based on their questionnaire and physical exam during screening. The officers underwent yearly fitness examinations as part of their professional responsibilities. While half the participants had a BMI that fell within the “overweight” range, their waist-to-thigh ratios — a reliable anthropometric measure of abdominal obesity—were within ranges for low diabetes and cardiovascular disease risk [86,87]. This is in contrast to previous reports of police officers having high rates of overweight and obesity, elevated blood pressure, and type 2 diabetes [57,58]. One factor that may explain this discrepancy is that we excluded participants who reported sleep disorders unrelated to their work schedule. In their study of North American police officers, Rajaratnam et al. [57] found that sleep disorders such as obstructive sleep apnea are highly comorbid with these health issues. Consequently, the dietary patterns and energy expenditure of this cohort may not reflect the behaviors of all shift-working police officers, who potentially eat different quantities of food at different times. Future research evaluating the efficacy of meal-timing interventions on the health of shift workers will require the inclusion of a broader cross-section of participants.

## 5. Conclusions

In summary, we characterized the dietary patterns and caloric intake of police officers across several typical days experienced in their rotating shift cycle. To our knowledge, this study is the first to do this by means of real-time meal-logging with timestamped photographs. In this manner, we could precisely assess how work schedules affected the timing of police officers’ meals and determine their macronutrient composition through dietetic analysis of the photographs. We observed that police officers ate significantly more on rest days and morning-shift days than evening- or night-shift days, including more calories from fat and saturated fat. However, the overall proportions of macronutrients in their diets did not substantially differ between days. Importantly, we observed a series of dietary patterns that implicate food timing—rather than its quantity or composition—as the differentiating nutritional factor hypothesized to affect metabolic health during different work shifts. Future studies are needed to test whether meal-timing interventions, including limiting night-time caloric intake and/or extending the fasting duration, are effective and feasible countermeasures to prevent the adverse metabolic consequences of shift work.

## Figures and Tables

**Figure 1 nutrients-12-00999-f001:**
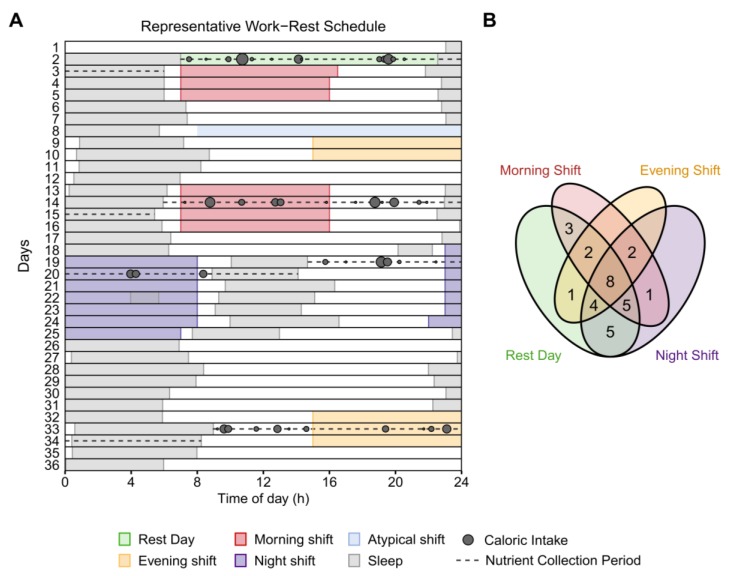
Raster plot of the study schedule and Venn diagram of day type contributions. (**A**) Diagram of the work–rest schedule, sleep–wake patterns, and eating behavior of one participant on a 35-day cycle. Successive study days are depicted on the vertical axis and time-of-day along the abscissa. Grey rectangles depict sleep episodes and colored rectangles represent shifts. Dashed horizontal lines identify the relevant nutrient collection periods for each “day type” (from top to bottom: rest, morning-shift, night-shift, and evening-shift days)—beginning from the wake time of the main sleep period prior to each shift/rest day until the wake time of the following main sleep period. Grey circles represent caloric intakes of different sizes. (**B**) Venn diagram of the number of participants contributing to each combination of day types. Of the 31 participants, 8 contributed to all 4 possible day types, 21 contributed to at least 3 day types, and all 31 contributed to at least 2 day types.

**Figure 2 nutrients-12-00999-f002:**
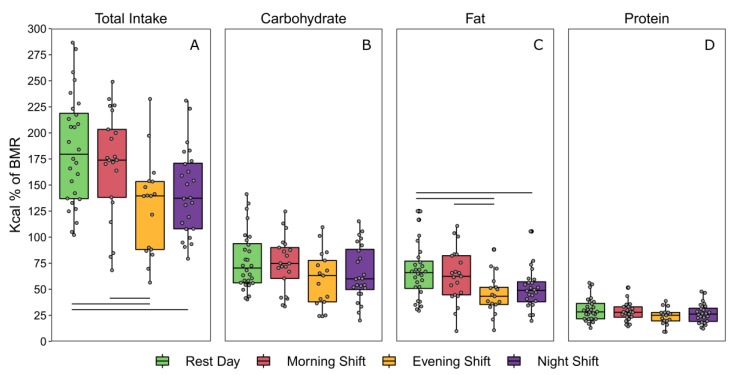
Total and macronutrient caloric intake across each day type (adjusted for 24-h days) expressed as percentages of basal metabolic rate (% BMR). Boxplots depict the median and interquartile ranges of total % BMR from (**A**) all calories, (**B**) carbohydrates, (**C**) fats, and (**D**) proteins. Horizontal lines represent significant differences (adjusted, *p* < 0.05) between day types relative to metabolic requirements.

**Figure 3 nutrients-12-00999-f003:**
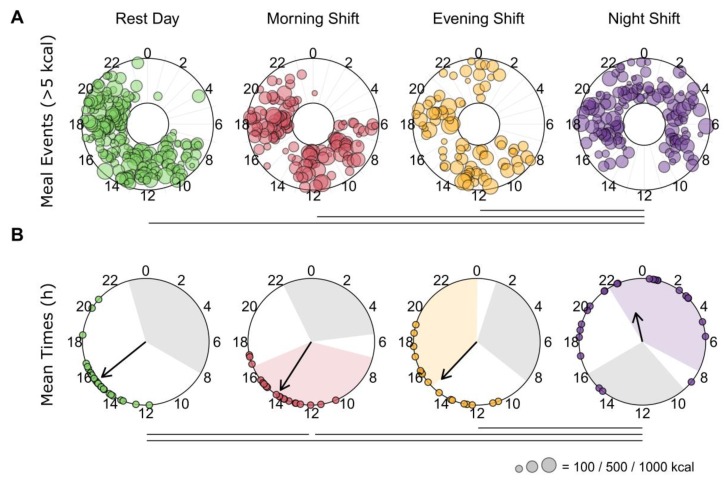
Meal events and average caloric intake by time-of-day. (**A**) Individual meal intakes (>5 kcal) are depicted as colored circles in concentric rings for each participant. The sizes of the colored circles reflect the total caloric content of the meal. Horizontal lines represent significant differences in the spread or dispersion of caloric intake around the clock between day types. The spread per participant is derived from rho, the mean resultant vector length of circular data. On average, calories were more dispersed for night-shift days than other day types (Appendix A). (**B**) Weighted circular mean times of caloric intake for each participant on each day type are depicted as colored circles. The direction of the arrows identifies the group circular mean time for each day type. The length of the arrows represents the consistency of individual mean intake times, with smaller arrows indicating a larger spread. Colored regions depict the average work hours for each shift type. The grey regions depict the average times of the main sleep period for each shift type (based on the wake and sleep onset times before and after each day, respectively). Horizontal lines represent significant differences (adjusted, *p* < 0.05) in the timing of caloric intake between day types. Mean times of caloric intake occur later on night-shift days than all other days, and earlier on morning-shift days than rest days (Appendix A).

**Figure 4 nutrients-12-00999-f004:**
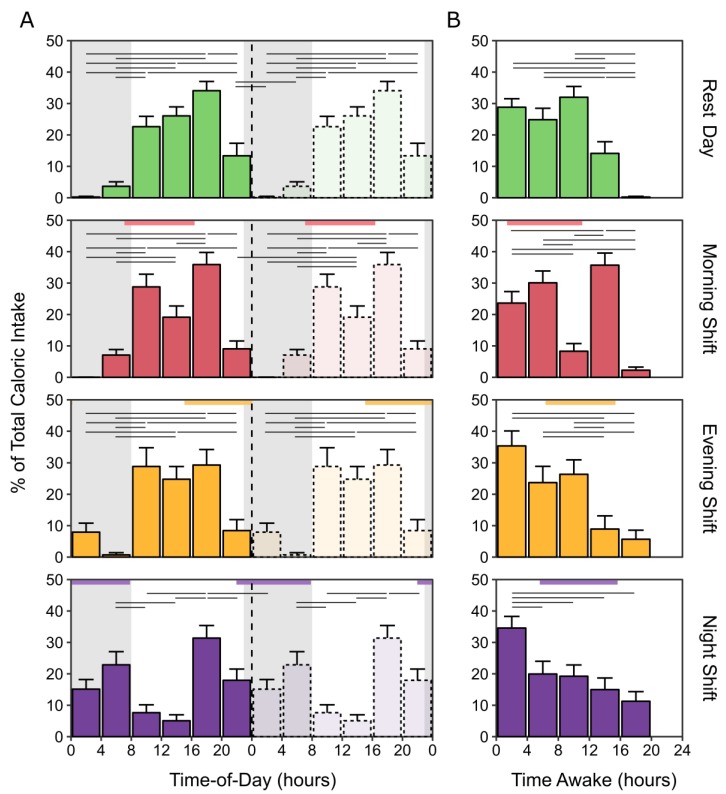
Distribution of calories by time-of-day (double-plotted) and time awake per day type. Percentages of total caloric intake on rest days, morning-shift days, evening-shift days, and night-shift days are depicted along the abscissa in 4-h bins by (**A**) time-of-day, replotted on the right, and (**B**) time awake. Grey regions represent the average sleep periods on rest days. Colored horizontal rectangles indicate the average work start and end times for each shift type. Horizontal black lines indicate significant differences (adjusted, *p* < 0.05) in the proportion of total calories consumed between time-of-day or time awake bins. Error bars represent the standard error of the mean.

**Table 1 nutrients-12-00999-t001:** Participants’ characteristics overall and by sex.

	Total (*n* = 31)		Male (*n* = 25)		Female (*n* = 6)		Welch’s T-test ^1^	
Characteristic	M	SD	M	SD	M	SD	*t* (29)	*P*
Age (years)	32.1	5.4	32.2	5.7	32.0	4.0	0.10	0.921
BMI (kg/m^2^)	25.0	2.3	25.3	2.2	23.6	2.1	1.79	0.260
WTR	1.68	0.19	1.69	0.19	1.66	0.21	0.32	0.887
WHR	0.898	0.073	0.911	0.069	0.846	0.071	2.01	0.286
BMR (kcal)	1706	197	1782	121	1388	109	7.77	<0.001
MSF (hh:mm)	03:02	01:20	02:56	01:19	03:26	01:29	0.71	0.709
Seniority (years)	8.6	4.7	8.9	5.0	7.3	3.4	0.97	0.617

^1^ Welch’s T-test for Unequal Variances testing for significant differences (Benjamini–Hochberg adjusted, *p* < 0.05). Body mass index (BMI); waist-to-thigh ratio (WTR); waist-to-hip ratio (WHR); estimated basal metabolic rate (BMR) = energy (kcal) required to maintain metabolic function at rest for 24 h [59]; chronotype based on self-reported mid-sleep on free days (MSF) after evening shifts [72]; years in the police force (Seniority).

**Table 2 nutrients-12-00999-t002:** Estimated basal metabolic rate (BMR), meal count, and total overall and macronutrient caloric intake expressed as percentages of BMR across each day type.

	Rest Day (*n* = 28)		Morning Shift (*n* = 21)		Evening Shift (*n* = 17)		Night Shift (*n* = 24)		Effect of Day Type ^3^	
	M	SD	M	SD	M	SD	M	SD	*F* (df)	*p*
24-h BMR (kcal) ^1^	1708	191	1734	210	1717	229	1679	205	0.81 (3,84)	0.491
Meal count (n)	7.5 ^M,E,N^	2.2	6.3 ^R,E,N^	2.1	5.0 ^R,M,N^	1.4	5.9 ^R,M,E^	1.7	11.91 (3,58)	<0.001
Total Intake (%) ^2^	183.5 ^E,N^	52.8	170.5 ^E^	51.4	129.6 ^R,M^	46.4	142.7 ^R^	41.6	7.90 (3,61)	0.001
Carbohydrate (%)	77.0	28.1	74.5	25.8	59.6	26.8	65.4	26.4	2.95 (3,59)	0.064
Sugars (%)	29.2	14.5	32.4	14.1	25.8	12.9	26.6	13.7	1.09 (3,61)	0.411
Total Fat (%)	67.4 ^E,N^	26.9	62.8 ^E^	25.9	45.2 ^R,M^	19.3	50.2 ^R^	18.8	5.67 (3,63)	0.004
Saturated Fat (%)	23.4 ^E,N^	12.4	20.0	9.9	15.2 ^R^	7.9	16.6 ^R^	7.6	4.85 (3,63)	0.008
Total Protein (%)	30.4	11.1	28.2	9.0	23.7	8.2	26.8	9.5	2.02 (3,60)	0.161

^1^ Caloric intake (kcal) required to maintain metabolic function at rest for 24 h [59]. ^2^ Total intake as a percentage of BMR includes calories from carbohydrate, fat, protein, and alcohol. ^3^ Mixed-effects models compared the effect of day type, corrected for sex and age; *p*-values adjusted with the Benjamini–Hochberg method. Percentage of estimated basal metabolic rate (% BMR) weighted for day length, i.e., from wake time of the main sleep preceding the shift/rest day until the wake time of the one following; ^R,M,E,N^ = significant differences (adjusted, *p* < 0.05) between corresponding day types (i.e., Rest, Morning, Evening, Night).

**Table 3 nutrients-12-00999-t003:** Percentages of total caloric intake per day type at night and at work.

	Rest Day (*n* = 28)		Morning Shift (*n* = 21)		Evening Shift (*n* = 17)		Night Shift (*n* = 24)		Effect of Day Type ^4^	
Nutrient Intake	M	SD	M	SD	M	SD	M	SD	*F* (df)	*p*
**Overall** ^1^										
Total Carbohydrate (%)	42.2	9.8	44.5	10.2	45.2	10.1	45.2	10.5	0.52 (3,58)	0.986
All Sugars (%)	15.8	6.2	19.1	5.9	20.1	8.2	18.5	8.1	1.41 (3,57)	0.620
Total Fat (%)	36.1	7.2	36.1	9.2	35.0	9.8	35.3	8.4	0.05 (3,63)	0.986
Saturated Fat (%)	12.2	3.7	11.4	4.0	11.8	4.2	11.6	3.8	0.26 (3,65)	0.986
Total Protein (%)	17.0	5.5	17.1	4.0	19.0	6.0	19.2	5.8	2.04 (3,59)	0.591
**At Night** ^2^										
Total Intake (%)	2.2 ^E,N^	7.4	1.2 ^E,N^	3.0	9.7 ^R,M,N^	11.8	30.3 ^R,M,E^	16.7	35.70 (3,84)	<0.001
Total Carbohydrate (%)	1.3 ^N^	4.7	0.4 ^N^	1.6	4.5 ^N^	5.4	13.5 ^R,M,E^	8.7	24.54 (3,84)	<0.001
All Sugars (%)	0.8 ^N^	2.8	0.3 ^N^	1.2	1.6 ^N^	2.9	5.8 ^R,M,E^	4.6	14.47 (3,84)	<0.001
Total Fat (%)	0.7 ^E,N^	2.1	0.6 ^E,N^	1.8	3.3 ^R,M,N^	4.9	10.7 ^R,M,E^	6.9	28.41 (3,84)	<0.001
Saturated Fat (%)	0.1 ^E,N^	0.4	0.1 ^E,N^	0.4	1.2 ^R,M,N^	1.9	3.1 ^R,M,E^	2.2	22.71 (3,68)	<0.001
Total Protein (%)	0.2 ^N^	0.7	0.2^N^	0.6	1.2 ^N^	1.6	6.0 ^R,M,E^	4.0	38.26 (3,62)	<0.001
**At Work** ^3^										
Total Intake (%)	-	-	52.5	14.3	39.0	19.7	43.4	17.4	4.07 (2,42)	0.083
Total Carbohydrate (%)	-	-	24.2	7.8	15.8	8.7	19.9	9.9	3.86 (2,43)	0.083
All Sugars (%)	-	-	9.9	4.3	7.5	5.3	8.6	6.0	1.20 (2,36)	0.377
Total Fat (%)	-	-	19.5	6.8	14.9	11.7	15.3	7.7	3.47 (2,38)	0.083
Saturated Fat (%)	-	-	6.1	2.5	4.6	4.0	4.7	3.1	2.10 (2,41)	0.204
Total Protein (%)	-	-	8.8	3.4	7.9	5.1	8.2	4.0	0.56 (2,34)	0.576

^1^ Intake during the day from wake time preceding shift/rest day to wake time following. ^2^ Intake from 2300–0600 h. ^3^ Intake from work start to work end. ^4^ Mixed-effects model ANOVA comparing the effect of day type, corrected for sex and age; *P*-values for Overall, At Night, and At Work adjusted separately with the Benjamini–Hochberg method. ^R,M,E,N^ = significant differences (adjusted, *p* < 0.05) between corresponding day types (i.e., Rest, Morning, Evening, Night).

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
