# Peer review of "Effects of Shift Work on the Eating Behavior of Police Officers on Patrol"

_nutrients, 2020, doi:10.3390/nu12040999_

Round 1

Reviewer 1 Report

In this study, the authors aimed to characterize the dietary patterns and caloric intake of police officers across different rotating shift types during their work cycle by means of real-time meal-logging with time-stamped photographs using a mobile phone.

This is the first report to do this by means of real-time meal-logging with time-stamped photographs. The authors did a great job on this noteworthy issue. Therefore, I applaud the authors for their efforts.

I am interested in the topic reported. Nevertheless, the data is mainly collected from a specific community. Why did the authors choose police officers to be enrolled in their study? Can this small group represent and draw conclusions for all types of shift workers? It would be very interesting if the authors continue some information about other shift workers professionals, although they mentioned a sentence about “healthcare professionals” near the end of section 4.4. Some expansion with strengthening of the issue would be appreciated.

Although the authors have stated that a subset of 53 police officers from a larger study was invited to participate; 36 agreed to do so. Of these, 31 officers contributed to the final dataset as 5 had to be excluded due to dropout (n=1), technical issues with equipment (n=2) or withdrawal for medical reasons (n=2). Compliance of the participants to your study seemed high. Please indicate the selection of study participants by adding a CONSORT diagram with flow chart in your manuscript.

I encourage the authors to discuss some possible complications or side effects of shift work on the eating behavior of police officers.

Reviewer 2 Report

Overall, this manuscript is excellent, and worthy of publication. Several aspects of the study design (in particular rostering) remain unclear however. Please see my comments and concerns listed below.

Include in the review of the literature work specific to police officers. This paper currently reads like cops were a convenient sample vs. a target population with a specific problem.

The rostering schedule needs some clarification – were some officers assigned to 8 hour shifts and some to 12 hour shifts? If so, how many officers in each group? Were the shifts rapid forward rotation (day, evening, night)? It appears from figure 1a that this might not be the case…? What was the extent of overtime?

How was compliance among officers with wearing the bands and filling out diaries as well as taking food pictures?

Were notes submitted in cases where photographs of food may have been ambiguous? E.g. a plant based vs. a beef burger or a vegetarian vs. a meat chili?

Page 5, line 195, I assume it should be figure 1B (not S1B)?

The roster is clearly complex, and I appreciate the effort to clarify with figures 1A and B, but confusion remains. For example, it’s unclear where the rest is represented within figure 1A. Also, it appears from figure 1B that three participants did not have rest days? It is unclear how many consecutive days participants were monitored for.

Why were no other sleep metrics considered, such as efficiency, latency, onset and wake variability from actigraphs or subjective measures such as sleepiness from diaries?

Did any interrater reliability testing occur on photo analysis?

What is table S1 (page 7, line 284)? Table 1 does not appear to contain shift start and end times.

Figure 2 is much clearer!

Again, unclear what table S2 and figure S3 are…

Within the discussion, tie back to the importance of this investigation for police officers, given their rates of obesity, diabetes, etc.

I see upon conclusion of review that S tables and figures are supplementary – please state as such in the body of the text to avoid reader confusion.
